# CYCLADES:
# Conflict-free Asynchronous Machine Learning

**Xinghao Pan,**[*] **Maximilian Lam,**[*] **Stephen Tu,**[*] **Dimitris Papailiopoulos,**[*]
**Ce Zhang,**[†] **Michael I. Jordan**[*‡], **Kannan Ramchandran,**[*] **Chris Re,**[†] **Benjamin Recht**[*‡]

## Abstract

We present CYCLADES, a general framework for parallelizing stochastic optimization algorithms in a shared memory setting. CYCLADES is asynchronous during model updates, and requires no memory locking mechanisms, similar to HOGWILD!-type algorithms. Unlike HOGWILD!, CYCLADES introduces no conflicts during parallel execution, and offers a black-box analysis for provable speedups across a large family of algorithms. Due to its inherent cache locality and conflict-free nature, our multi-core implementation of CYCLADES consistently outperforms HOGWILD!-type algorithms on sufficiently sparse datasets, leading to up to $40\%$ speedup gains compared to HOGWILD!, and up to $5\times$ gains over asynchronous implementations of variance reduction algorithms.

## 1 Introduction

Following the seminal work of HOGWILD! [17], many studies have demonstrated that near linear speedups are achievable on several machine learning tasks via asynchronous, lock-free implementations [25, 13, 8, 16]. In all of these studies, classic algorithms are parallelized by simply running parallel and asynchronous model updates without locks. These lock-free, asynchronous algorithms exhibit speedups even when applied to large, non-convex problems, as demonstrated by deep learning systems such as Google's Downpour SGD [6] and Microsoft's Project Adam [4]. While these techniques have been remarkably successful, many of the above papers require delicate and tailored analyses to quantify the benefits of asynchrony for each particular learning task. Moreover, in non-convex settings, we currently have little quantitative insight into how much speedup is gained from asynchrony.

In this work, we present CYCLADES, a general framework for lock-free, asynchronous machine learning algorithms that obviates the need for specialized analyses. CYCLADES runs asynchronously and *maintains serial equivalence*, i.e., it produces the same outcome as the serial algorithm. Since it returns the same output as a serial implementation, any algorithm parallelized by our framework inherits the correctness proof of the serial counterpart without modifications. Additionally, if a particular serially run heuristic is popular, but does not have a rigorous analysis, CYCLADES still guarantees that its execution will return a serially equivalent output.

CYCLADES achieves serial equivalence by partitioning updates among cores, in a way that ensures that there are no conflicts across partitions. Such a partition can always be found efficiently by leveraging a powerful result on graph phase transitions [12]. When applied to our setting, this result guarantees that a sufficiently small sample of updates will have only a *logarithmic* number of conflicts. This allows us to evenly partition model updates across cores, with the guarantee that all conflicts are localized within each core. Given enough problem sparsity, CYCLADES guarantees a nearly linear

---

[*]Department of Electrical Engineering and Computer Science, UC Berkeley, Berkeley, CA.

[†]Department of Computer Science, Stanford University, Palo Alto, CA.

[‡]Department of Statistics, UC Berkeley, Berkeley, CA.

speedup, while inheriting all the qualitative properties of the serial counterpart of the algorithm, e.g., proofs for rates of convergence. Enforcing a serially equivalent execution in CYCLADES comes with additional practical benefits. Serial equivalence is helpful for hyperparameter tuning, or locating the best model produced by the asynchronous execution, since experiments are reproducible, and solutions are easily verifiable. Moreover, a CYCLADES program is easy to debug because bugs are repeatable and we can examine the step-wise execution to localize them.

A significant benefit of the update partitioning in CYCLADES is that it induces considerable access locality compared to the more unstructured nature of the memory accesses during HOGWILD!. Cores will access the same data points and read/write the same subset of model variables. This has the additional benefit of reducing false sharing across cores. Because of these gains, CYCLADES can actually *outperform* HOGWILD! in practice on sufficiently sparse problems, despite appearing to require more computational overhead. Remarkably, because of the added locality, even a single threaded implementation of CYCLADES can actually be faster than serial SGD. In our SGD experiments for matrix completion and word embedding problems, CYCLADES can offer a speedup gain of up to $40\%$ compared to that of HOGWILD!. Furthermore, for variance reduction techniques such as SAGA [7] and SVRG [11], CYCLADES yields better accuracy and more significant speedups, with up to $5\times$ performance gains over HOGWILD!-type implementations.

## 2 The Algorithmic Family of Stochastic-Updates

We study parallel asynchronous iterative algorithms on the computational model used by [17]: several cores have access to the same shared memory, and each of them can read and update components of the shared memory. In this work, we consider a family of randomized algorithms that we refer to as *Stochastic Updates* (SU). The main algorithmic component of SU focuses on updating small subsets of a model variable $\mathbf{x}$, according to prefixed access patterns, as sketched by Alg. 1.

In Alg. 1, $\mathcal{S}_i$ is a subset of the coordinates in $\mathbf{x}$, each function $f_i$ operates on the subset $\mathcal{S}_i$ of coordinates, and $u_i$ is a local update function that computes a vector with support on $\mathcal{S}_i$ using as input $\mathbf{x}_{\mathcal{S}_i}$ and $f_i$. Moreover, $T$ is the total number of iterations, and $\mathcal{D}$ is the distribution with support $\{1, \ldots, n\}$ from which we draw $i$. Several machine learning algorithms belong to the SU algorithmic family, such as stochastic gradient descent (SGD), with or without weight decay and regularization, variance-reduced

---

**Algorithm 1** Stochastic Updates

1: Input: $\mathbf{x}$; $f_1, \ldots, f_n$; $T$
2: **for** $t = 1 : T$ **do**
3:     sample $i \sim \mathcal{D}$
4:     $\mathbf{x}_{\mathcal{S}_i} = u_i(\mathbf{x}_{\mathcal{S}_i}, f_i)$
5: **Output: x**

---

learning algorithms like SAGA and SVRG, and even some combinatorial graph algorithms. In our supplemental material, we explain how these algorithms can be phrased in the SU language.

**The updates conflict graph**    A useful construct for our developments is the conflict graph between updates, which can be generated from the bipartite graph between the updates and the model variables. We define these graphs below, and provide an illustrative sketch in Fig. 1.

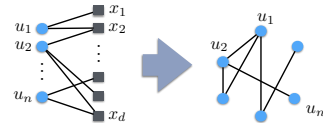

**Definition 1.** *Let $G_u$ denote the bipartite update-variable graph between the $n$ updates and the $d$ model variables. An update $u_i$ is linked to a variable $x_j$, if $u_i$ requires to read/write $x_j$. Let $E_u$ denote the number of edges in the bipartite graph, $\Delta_L$ the max left degree of $G_u$, and $\overline{\Delta}_L$ the average left degree. Finally, we denote by $G_c$ the conflict graph on the $n$ updates. Two vertices in $G_c$ are linked, if the corresponding updates share at least one variable in $G_u$. We also denote as $\Delta$ the max vertex degree of $G_c$.*

Figure 1: In the bipartite graph, an update $u_i$ is linked to variable $x_j$ when it needs to read/write it. From $G_u$ we obtain the conflict graph $G_c$, whose max degree is $\Delta$. If that is small, we expect that it is possible to parallelize updates without too many conflicts. CYCLADES exploits this intuition.

We stress that the conflict graph is never constructed, but is a useful for understanding CYCLADES.

**Our Main Result**    By exploiting the structure of the above graphs and through a light-weight sampling and allocation of updates, CYCLADES guarantees the following result for SU algorithms, which we establish in the following sections.

**Theorem 1** (informal)**.** *Let an SU algorithm $\mathcal{A}$ be defined through $n$ update rules, where the conflict max degree between the $n$ updates is $\Delta$, and the sampling distribution $\mathcal{D}$ is uniform with (or without) replacement from $\{1, \ldots, n\}$. Moreover, assume that we wish to run $\mathcal{A}$ for $T = \Theta(n)$ iterations, and*

*that $\frac{\Delta_L}{\overline{\Delta}_L} \leq \sqrt{n}$. Then on up to $P = \tilde{O}(\frac{n}{\Delta \cdot \Delta_L})$ cores, CYCLADES guarantees a $\widetilde{\Omega}(P)$ speedup over $\mathcal{A}$, while outputting the same solution $\mathbf{x}$ as $\mathcal{A}$ would do after the same random set of $T$ iterations.*[4]

We now provide two examples of how these guarantees translate for specific problem cases.

**Example 1.** *In many applications we seek to minimize: $\min_{\mathbf{x}} \frac{1}{n} \sum_{i=1}^{n} \ell_i(\mathbf{a}_i^T \mathbf{x})$ where $\mathbf{a}_i$ represents the $i$th data point, $\mathbf{x}$ is the parameter vector, and $\ell_i$ is a loss. Several problems can be formulated in this way, such as logistic regression, least squares, binary classification, etc. If we tackle the above problem using SGD, or techniques like SVRG and SAGA, then (as we show in the supplemental) the update sparsity is determined by the gradient of a single sampled data point $\mathbf{a}_i$. Here, we will have $\Delta_L = \max_i \|\mathbf{a}_i\|_0$, and $\Delta$ will be equal to the maximum number of data points $\mathbf{a}_i$ that share at least one feature. As a toy example, let $\frac{n}{d} = \Theta(1)$ and let the non-zero support of $\mathbf{a}_i$ be of size $n^\delta$ and uniformly distributed. Then, one can show that with high probability $\Delta = \widetilde{O}(n^{1/2+\delta})$ and hence CYCLADES achieves an $\widetilde{\Omega}(P)$ speedup on up to $P = \widetilde{O}(n^{1/2-2\delta})$ cores.*

**Example 2.** *Consider the generic optimization $\min_{\mathbf{x}_i, \mathbf{y}_j, i \in [n]} \sum_{i=1}^{m} \sum_{j=1}^{m} \phi_{i,j}(\mathbf{x}_i, \mathbf{y}_j)$, which captures several problems like matrix completion and factorization [17], word embeddings [2], graph $k$-way cuts [17], etc. Assume that we aim to minimize the above by sampling a single function $\phi_{i,j}$ and then updating $\mathbf{x}_i$ and $\mathbf{y}_j$ using SGD. Here, the number of update functions is proportional to $n = m^2$, and each gradient update with respect to the sampled function $\phi_{i,j}(\mathbf{x}_i, \mathbf{y}_j)$ is only interacting with the variables $\mathbf{x}_i$ and $\mathbf{y}_j$, i.e., only two variable vectors out of the $2m$ vectors (i.e., $\Delta_L = 2$). This also implies a conflict degree of at most $\Delta = 2m$. Here, CYCLADES can provably guarantee an $\widetilde{\Omega}(P)$ speedup for up to $P = O(m)$ cores.*

In our experiments we test CYCLADES on several problems including least squares, classification with logistic models, matrix factorization, and word embeddings, and several algorithms including SGD, SVRG, and SAGA. We show that in most cases it can significantly outperform the HOGWILD! implementation of these algorithms, if the data is sparse.

**Remark 1.** *We would like to note that there are several cases where there might be a few outlier updates with extremely high conflict degree. In the supplemental material, we prove that if there are no more than $O(n^\delta)$ vertices of high conflict degree $\Delta_o$, and the rest of the vertices have max degree at most $\Delta$, then the result of Theorem 1 still holds in expectation.*

In the following section, we establish the theory of CYCLADES and provide the details behind our parallelization framework.

## 3 CYCLADES: Shattering Dependencies

CYCLADES consists of three computational components as shown in Fig. 2. It starts by sampling (according to a distribution $\mathcal{D}$) a number of $B$ updates from the graph shown in Fig. 1, and assigns a label to each of them (a processing order). After sampling, it computes the connected components of the sampled subgraph induced by the $B$ sampled updates, to determine the conflict groups.

Once the conflicts groups are formed, it allocates them across $P$ cores. Finally, each core processes locally the conflict groups of updates that it has been assigned, following the order that each update has been labeled with. The above process is then repeated, for as many iterations as needed. The key component of CYCLADES is to carry out the sampling in such a way that we have as many connected components as possible, and all of them of small size, provably. In the next subsections, we explain how each part is carried out, and provide theoretical guarantees for each of them individually, which we combine at the end of this section for our main theorem.

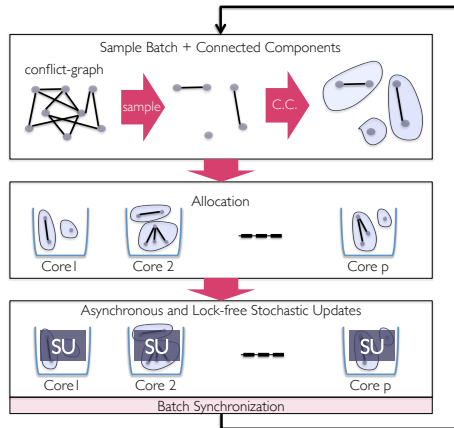

Figure 2: CYCLADES samples updates, finds conflict-groups, and allocates them across cores. Each core asynchronously updates the model, without access conflicts. This is possible by processing the conflicting updates within the same core.

A key technical aspect that we exploit in CYCLADES is that appropriate sampling and allocation of updates can lead to near optimal parallelization of SU algorithms. To do that we expand upon the following result established in [12].

**Theorem 2.** *Let $G$ be a graph on $n$ vertices, with max degree $\Delta$. Let us sample each vertex independently with probability $p = \frac{1-\epsilon}{\Delta}$ and define as $G'$ the induced subgraph on the sampled vertices. Then, the largest connected component of $G'$ has size at most $\frac{4}{\epsilon^2} \log n$, with high probability.*

The above result pays homage to the giant component phase transition phenomena in random Erdos-Renyi graphs. What is surprising is that similar phase transitions apply to any given graph!

In practice, for most SU algorithms of interest, the sampling distribution of updates is either with or without replacement from the $n$ updates. As it turns out, morphing Theorem 2 into a with-/without-replacement result is not straightforward. We defer the analysis needed to the supplemental material, and present our main theorem about graph sampling here.

**Theorem 3.** *Let $G$ be a graph on $n$ vertices, with max degree $\Delta$. Let us sample $B = \frac{(1-\epsilon)n}{\Delta}$ vertices with or without replacement, and define as $G'$ the induced subgraph on the sampled vertices. Then, the largest connected component of $G'$ has size at most $O(\frac{\log n}{\epsilon^2})$, with high probability.*

The key idea from the above is that if one samples no more than $B = (1-\epsilon)\frac{n}{\Delta}$ updates, then there will be at least $O\left(\epsilon^2 B / \log n\right)$ conflict groups to allocate across cores, each of size at most $O\left(\log n / \epsilon^2\right)$. Since there are no conflicts between different conflict-groups, the processing of updates per any single group will never interact with the variables corresponding to the updates of another conflict group. The next step of CYCLADES is to form and allocate the connected components (CCs) across cores, efficiently. We address this in the following subsection. In the following, for brevity we focus on the with-replacement sampling case, but the results can be extended to the without-replacement case.

**Identifying groups of conflict** In CYCLADES, we sample batches of updates of size $B$ multiple times, and for each batch we need to identify the conflict groups across the updates. Let us refer to $G_u^i$ as the subgraph induced by the $i$th sampled batch of updates on the update-variable graph $G_u$. In the following we always assume that we sample $n_b = c \cdot \Delta / (1 - \epsilon)$ batches, where $c \geq 1$ is a constant. This number of batches results in a constant number of passes over the dataset. Then, identifying the conflict groups in $G_u^i$ can be done with a connected components (CC) algorithm. The main question we need to address is what is the best way to parallelize this graph partitioning part. In the supplemental, we provide the details of this part, and prove the following result:

**Lemma 1.** *Let the number of cores be $P = O(\frac{n}{\Delta \Delta_L})$ and let $\frac{\Delta_L}{\Delta_L} \leq \sqrt{n}$. Then, the overall computation of CCs for $n_b = c \cdot \frac{\Delta}{1-\epsilon}$ batches, each of size $B = (1 - \epsilon)\frac{n}{\Delta}$, costs no more than $O(E_u/P \log^2 n)$.*

**Allocating updates to cores** Once we compute the CCs (i.e., the conflicts groups of the sampled updates), we have to allocate them across cores. Once a core has been assigned with CCs, it will process the updates included in these CCs, according to the order that each update has been labeled with. Due to Theorem 3, each connected component will contain at most $\mathcal{O}(\frac{\log n}{\epsilon^2})$ updates. Assuming that the cost of the $j$-th update in the batch is $w_j$, the cost of a single connected component $\mathcal{C}$ will be $w_\mathcal{C} = \sum_{j \in \mathcal{C}} w_j$. To proceed with characterizing the maximum load among the $P$ cores, we assume that the cost of a single update $u_i$, for $i \in \{1, \ldots, n\}$, is proportional to the out-degree of that update —according to the update-variable graph $G_u$— times a constant cost which we shall refer to as $\kappa$. Hence, $w_j = O(d_{L,j} \cdot \kappa)$, where $d_{L,j}$ is the degree of the $j$-th left vertex of $G_u$. In the supplemental material, we establish that a near-uniform allocation of CCs according to their weights leads to the following guarantee.

**Lemma 2.** *Let the number of cores by bounded as $P = O(\frac{n}{\Delta \Delta_L})$, and let $\frac{\Delta_L}{\Delta_L} \leq \sqrt{n}$. Then, computing the stochastic updates across all $n_b = c \cdot \frac{\Delta}{1-\epsilon}$ batches can be performed in time $\mathcal{O}(\frac{E \log^2 n}{P} \cdot \kappa)$, with high probability, where $\kappa$ is the per edge cost for computing one of the $n$ updates defined on $G_u$.*

**Stitching the pieces together** Now that we have described the sampling, conflict computation, and allocation strategies, we are ready to put all the pieces together and detail CYCLADES in full. Let us assume that we sample a total number of $n_b = c \cdot \frac{\Delta}{1-\epsilon}$ batches of size $B = (1 - \epsilon)\frac{n}{\Delta}$, and that each update is sampled uniformly at random. For the $i$-th batch let us denote as $\mathcal{C}_1^i, \ldots \mathcal{C}_{m_i}^i$ the connected

components on the induced subgraph $G_u^i$. Due to Theorem 3, each connected component $\mathcal{C}$ contains a number of at most $O(\frac{\log n}{\epsilon^2})$ updates; each update carries an ID (the order of which it would have been sampled by the serial algorithm). Using the above notation, we give the pseudocode for CYCLADES in Alg. 2. Note that the inner loop that is parallelized (i.e., the SU processing loop in lines $6-9$), can be performed asynchronously; cores do not have to synchronize, and do not need to lock any memory variables, as they are all accessing non-overlapping subset of $\mathbf{x}$. This also provides for better cache coherence. Moreover, each core potentially accesses the same coordinates several times, leading to good cache locality. These improved cache locality and coherence properties experimentally lead to substantial performance gains as we see in the next section. We can now combine the results of the previous subsection to obtain our main theorem for CYCLADES.

**Theorem 4.** *Let us assume any given update-variable graph $G_u$ with $\overline{\Delta}_L$ and $\Delta_L$, such that $\frac{\overline{\Delta}_L}{\Delta_L} \le \sqrt{n}$, and with induced max conflict degree $\Delta$. Then, CYCLADES on $P = O(\frac{n}{\Delta \cdot \Delta_L})$ cores, with batch sizes $B = (1-\epsilon)\frac{n}{\Delta}$ can execute $T = c \cdot n$ updates, for any constant $c \ge 1$, selected uniformly at random with replacement, in time $\mathcal{O}\left(\frac{E_u \cdot \kappa}{P} \cdot \log^2 n\right)$, with high probability.*

Observe that CYCLADES bypasses the need to establish convergence guarantees for the parallel algorithm. Hence, it could be the case for an applications of interest that we cannot analyze how a serial SU algorithm performs in terms of say the accuracy of the solution, but CYCLADES can still provide black box guarantees for speedup, since our analysis is completely oblivious to the qualitative performance of the serial algorithm. This is in contrast to recent studies similar to [5], where the authors provide speedup guarantees via a convergence-to-optimal proof for an asynchronous SGD on a nonconvex problem. Unfortunately these proofs can become complicated on a wider range of nonconvex objectives.

---

**Algorithm 2** CYCLADES

1: **Input:** $G_u, n_b$.
2: Sample $n_b$ subgraphs $G_u^1, \dots, G_u^{n_b}$ from $G_u$
3: Compute in parallel CCs for sampled graphs
4: **for** batch $i = 1 : n_b$ **do**
5:     Allocation of $\mathcal{C}_1^i, \dots \mathcal{C}_{m_i}^i$ to $P$ cores
6:     **for** each core **in parallel do**
7:         **for** each allocated component $\mathcal{C}$ **do**
8:             **for** each ordered update $j$ from $\mathcal{C}$ **do**
9:                 $\mathbf{x}_{\mathcal{S}_j} = u_j(\mathbf{x}_{\mathcal{S}_j}, f_j)$
10: **Output: x**

---

In the following section we show that CYCLADES is not only useful theoretically, but can consistently outperform HOGWILD! on sufficiently sparse datasets.

## 4 Evaluation

We implemented CYCLADES[5] in C++ and tested it on a variety of problems, and a number of stochastic updates algorithms, and compared against their HOGWILD! (i.e., asynchronous, lock-free) implementations. Since CYCLADES is intended to be a general SU parallelization framework, we do not compare against algorithms tailored to specific applications, nor do we expect CYCLADES to outperform every such highly-tuned, well-designed, specific algorithms. Our experiments were conducted on a machine with 72 CPUs (Intel(R) Xeon(R) CPU E7-8870 v3, 2.10 GHz) on 4 NUMA nodes, each with 18 CPUs, and 1TB of memory. We ran CYCLADES and HOGWILD! with 1, 4, 8, 16 and 18 threads pinned to CPUs on a single NUMA node (i.e., the maximum physical number of cores per single node), to can avoid well-known cache coherence and scaling issues across nodes [24].

| Dataset | # datapoints | # features | av. sparsity / datapoint | Comments |
|---------|--------------|------------|--------------------------|----------|
| NH2010 | 48,838 | 48,838 | 4.8026 | Topological graph |
| DBLP | 5,425,964 | 5,425,964 | 3.1880 | Authorship network |
| MovieLens | $\sim$10M | 82,250 | 200 | 10M movie ratings |
| EN-Wiki | 20,207,156 | 213,272 | 200 | Subset of english Wikipedia dump. |

Table 1: Details of datasets used in our experiments.

In our experiments, we measure overall running times which include the overheads for computing connected components and allocating work in CYCLADES. We also compute the objective value at the end of each epoch (i.e., one full pass over the data). We measure the speedups for each algorithm as $\frac{\text{time of the parallel algorithm to reach } \epsilon \text{ objective}}{\text{time of the serial algorithm to reach } \epsilon \text{ objective}}$ where $\epsilon$ was chosen to be the smallest objective value that is achievable by all parallel algorithms on every choice of number of threads. The serial algorithm used for comparison is HOGWILD! running serially on one thread. In Table 1 we list some details of the datasets that we use in our experiments. We tune our constant stepsizes so to maximize convergence

without diverging, and use one random data reshuffling across all epochs. Batch sizes are picked to optimize performance for CYCLADES.

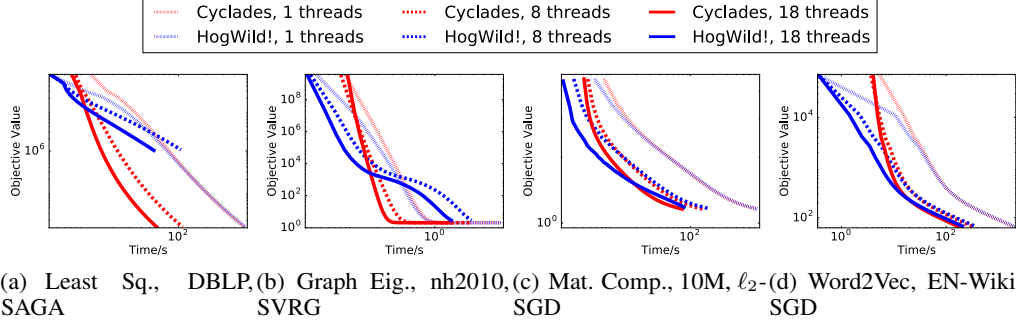

(a) Least Sq., DBLP, SAGA     (b) Graph Eig., nh2010, SVRG     (c) Mat. Comp., 10M, $\ell_2$-SGD     (d) Word2Vec, EN-Wiki, SGD

Figure 3: Convergence of CYCLADES and HOGWILD! in terms of overall running time with 1, 8, 16, 18 threads. CYCLADES is initially slower, but ultimately reaches convergence faster than HOGWILD!.

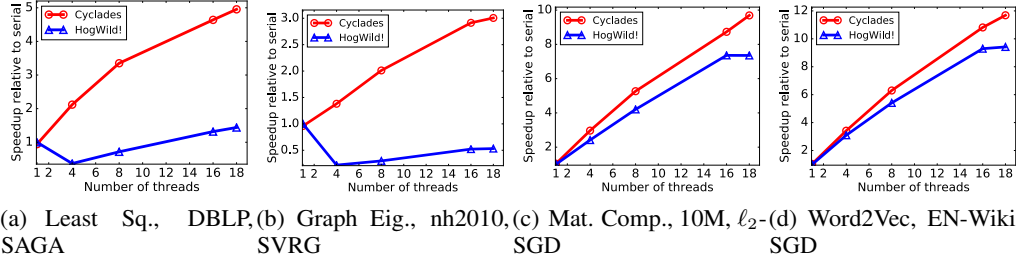

(a) Least Sq., DBLP, SAGA     (b) Graph Eig., nh2010, SVRG     (c) Mat. Comp., 10M, $\ell_2$-SGD     (d) Word2Vec, EN-Wiki, SGD

Figure 4: Speedup of CYCLADES and HOGWILD! versus number of threads. On multiple threads, CYCLADES always reaches $\epsilon$ objective faster than HOGWILD!. In some cases CYCLADES is faster than HOGWILD! even on 1 thread, due to better cache locality. In Figs. 4(a) and 4(b), CYCLADES exhibits significant gains since HOGWILD! suffers from asynchrony noise, and we had to use comparatively smaller stepsizes to prevent it from diverging.

**Least squares via SAGA** The first problem we consider is least squares: $\min_{\mathbf{x}} \min_{\mathbf{x}} \frac{1}{n} \sum_{i=1}^{n} (\mathbf{a}_i^T \mathbf{x} - b_i)^2$ which we will solve using the SAGA algorithm [7], an incrimental gradient algorithm with faster than SGD rates on convex, or strongly convex functions. In SAGA, we initialize $\mathbf{g}_i = \nabla f_i(\mathbf{x}_0)$ and iterate the following two steps $\mathbf{x}_{k+1} = \mathbf{x}_k - \gamma \cdot (\nabla f_{s_k}(\mathbf{x}_k) - \mathbf{g}_{s_k} + \frac{1}{n} \sum_{i=1}^{n} \mathbf{g}_i)$ and $\mathbf{g}_{s_k} = \nabla f_{s_k}(\mathbf{x}_k)$, where $f_i(\mathbf{x}) = (\mathbf{a}_i^T \mathbf{x} - b_i)^2$. In the above iteration it is useful to observe that the updates can be performed in a sparse and "lazy" way, as we explain in detail in our supplemental material.

The stepsizes chosen for each of CYCLADES and HOGWILD! were largest such that the algorithms did not diverge. We used the DBLP and NH2010 datasets for this experiment, and set $\mathbf{A}$ as the adjacency matrix of each graph. For NH2010, the values of $\mathbf{b}$ were set to population living in the Census Block. For DBLP we used synthetic values: we set $\mathbf{b} = \mathbf{A}\tilde{\mathbf{x}} + 0.1\tilde{\mathbf{z}}$, where $\tilde{\mathbf{x}}$ and $\tilde{\mathbf{z}}$ were generated randomly. The SAGA algorithm was run for 500 epochs for each dataset. When running SAGA for least squares, we found that HOGWILD! was divergent with the large stepsizes that we were using for CYCLADES (Fig. 5). Thus, in the multi-thread setting, we were only able to use smaller stepsizes for HOGWILD!, which resulted in slower convergence than CYCLADES, as seen in Fig. 3(a). The effects of a smaller stepsize for HOGWILD! are also manifested in terms of speedups in Fig. 4(a), since HOGWILD! takes a longer time to converge to an $\epsilon$ objective value.

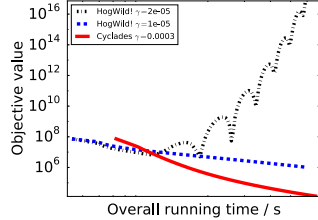

Figure 5: Convergence of CYCLADES and HOGWILD! on least squares using SAGA, with 16 threads, on DBLP dataset. HOGWILD! diverges with $\gamma > 10^{-5}$; thus, we were only able to use a smaller step size $\gamma = 10^{-5}$ for HOGWILD! on multiple threads. For HOGWILD! on 1 thread (and CYCLADES on any number of threads), we could use a larger stepsize of $\gamma = 3 \times 10^{-4}$.

**Graph eigenvector via SVRG** Given an adjacency matrix $\mathbf{A}$, the top eigenvector of $\mathbf{A}^T \mathbf{A}$ is useful in several applications such as spectral clustering, principle component analysis, and others. In a

recent work, [10] proposes an algorithm for computing the top eigenvector of $\mathbf{A}^T\mathbf{A}$ by running intermediate SVRG steps to approximate the shift-and-invert iteration. Specifically, at each step SVRG is used to solve: $\min \sum_{i=1}^{n}\left(\frac{1}{2}\mathbf{x}^T\left(\frac{\lambda}{n}\mathbf{I} - \mathbf{a}_i\mathbf{a}_i^T\right)\mathbf{x} - \frac{1}{n}\mathbf{b}^T\mathbf{x}\right)$, where $\mathbf{a}_i$ is the $i$-th column of $\mathbf{A}$. According to [10], if we initialize $\mathbf{y} = \mathbf{x}_0$ and assume $\|\mathbf{a}_i\| = 1$, we have to iterate the following updates $\mathbf{x}_{k+1} = \mathbf{x}_k - \gamma \cdot n \cdot (\nabla f_{s_k}(\mathbf{x}_k) - \nabla f_{s_k}(\mathbf{y})) + \gamma \cdot \nabla f(\mathbf{y})$ where after every $T$ iterations we update $\mathbf{y} = \mathbf{x}_k$, and the stochastic gradients are of the form $\nabla f_i(\mathbf{x}) = \left(\frac{\lambda}{n}\mathbf{I} - \mathbf{a}_i\mathbf{a}_i^T\right)\mathbf{x} - \frac{1}{n}\mathbf{b}$.

We apply CYCLADES to the above SVRG iteration (see supplemental) for parallelizing this problem. We run experiments on two graphs: DBLP and and NH2010. We ran SVRG for 50 and 100 epochs for NH2010 and DBLP respectively. The convergence of SVRG for graph eigenvectors is shown in Fig. 3(b). CYCLADES starts off slower than HOGWILD!, but always produces results equivalent to the convergence on a single thread. HOGWILD! does not exhibit the same behavior on multiple threads as it does serially; due to asynchrony causes HOGWILD! to converge slower on multiple threads. This effect is clearly seen on Figs. 4(b), where HOGWILD! fails to converge faster than the serial counterpart, and CYCLADES attains a significantly better speedup on 16 threads.

**Matrix completion and word embeddings via SGD** In matrix completion we are given a partially observed matrix $\mathbf{M}$, and wish to factorize it as $\mathbf{M} \approx \mathbf{UV}$ where $\mathbf{U}$ and $\mathbf{V}$ are low rank matrices with dimensions $n \times r$ and $r \times m$ respectively. This may be achieved by optimizing $\min \sum_{(i,j)\in\Omega}(M_{i,j} - \mathbf{U}_{i,\cdot}\mathbf{V}_{\cdot,j})^2 + \frac{\lambda}{2}(\|\mathbf{U}\|_F^2 + \|\mathbf{V}\|_F^2)$ where $\Omega$ is the set of observed entries, which can be approximated by SGD on the observed samples. The regularized objective can be optimized by weighted SGD. In our experiments, we chose a rank of $r = 100$, and ran SGD and weighted SGD for 200 epochs. We used the MovieLens 10M dataset containing 10M ratings for 10K movies by 72K users.

Our second task that uses SGD is word embeddings, which aim to represent the meaning of a word $w$ via a vector $\mathbf{v}_w \in \mathbb{R}^d$. A recent work by [2] proposes to solve: $\min_{\{\mathbf{v}_w\},C} \sum_{w,w'} A_{w,w'}(\log(A_{w,w'}) - \|\mathbf{v}_w + \mathbf{v}_{w'}\|_2^2 - C)^2$, where $A_{w,w'}$ is the number of times words $w$ and $w'$ co-occur within $\tau$ words in the corpus. In our experiments we set $\tau = 10$ following the suggested recipe of the aforementioned paper. We can approximate the solution to the above problem, by obtaining one using SGD: we can repeatedly sample entries $A_{w,w'}$ from $\mathbf{A}$ and update the corresponding vectors $\mathbf{v}_w, \mathbf{v}_{w'}$. Then, at the end of each full pass over the data, we update the constant $C$ by its locally optimal value, which can be calculated in closed form. In our experiments, we optimized for a word embedding of dimension $d = 100$, and tested on a 80MB subset of the English Wikipedia dump. For our experiments, we run SGD for 200 epochs.

Figs. 3(c) and 3(d) show the convergence for the matrix completion and word embeddings problems. CYCLADES is initially slower than HOGWILD! due to the overhead of computing connected components. However, due to better cache locality and convergence properties, CYCLADES is able to reach a lower objective value in less time than HOGWILD!. In fact, we observe that CYCLADES is faster than HOGWILD! when both are run serially, demonstrating that the gains from (temporal) cache locality outweigh the coordination overhead of CYCLADES. These results are reflected in the speedups of CYCLADES and HOGWILD! (Figs. 4(c) and 4(d)). CYCLADES consistently achieves a better speedup (up to $11\times$ on 18 threads) compared to that of HOGWILD! (up to $9\times$ on 18 threads).

**Partitioning and allocation costs**[5] The cost of partitioning and allocation[5] for CYCLADES is given in Table 2, relatively to the time that HOGWILD! takes to complete a single pass over the dataset. For matrix completion and the graph eigenvector problem, on 18 threads, CYCLADES takes the equivalent of 4-6 epochs of HOGWILD! to complete its partitioning, as the problem is either very sparse or the updates are expensive. For solving least squares using SAGA and word embeddings using SGD, the cost of partitioning is equivalent to 11-14 epochs of HOGWILD! on 18 threads. However, we point out that partitioning and allocation[5] is a one-time cost which becomes cheaper with more stochastic update epochs. Additionally, note that this cost can become amortized due to the extra experiments one has to run for hyperparameter tuning, since the graph partitioning is identical across different stepsizes one might want to test.

**Binary classification and dense coordinates** Here we explore settings where CYCLADES is expected to perform poorly due to the inherent density of updates (i.e., for data sets with dense features). In particular, we test CYCLADES on a classification problem for text based data. Specifically, we run classification for the URL dataset [15] contains $\sim 2.4\text{M}$ URLs, labeled as either benign or malicious,

| # threads | Least Squares SAGA, DBLP | Graph Eig. SVRG, NH2010 | Mat. Comp. $\ell_2$-SGD, MovieLens | Word2Vec SGD, EN-Wiki |
|---|---|---|---|---|
| 1 | 2.2245 | 0.9039 | 0.5507 | 0.5299 |
| 18 | 14.1792 | 4.7639 | 5.5270 | 3.9362 |

Table 2: Ratio of the time that CYCLADES consumes for partition and allocation over the time that HOGWILD! takes for 1 full pass over the dataset. On 18 threads, CYCLADES takes between 4-14 HOGWILD! epochs to perform partitioning. Note however, this computational effort is only required once per dataset.

and 3.2M features, including bag-of-words representation of tokens in the URL. For this classification task, we used a logistic regression model, trained using SGD. By its power-law nature, the dataset consists of a small number of extremely dense features which occur in nearly all updates. Since CYCLADES explicitly avoids conflicts, it has a schedule of SGD updates that leads to poor speedups.

However, we observe that most conflicts are caused by a small percentage of the densest features. If these features are removed from the dataset, CYCLADES is able to obtain much better speedups. The speedups that are obtained by CYCLADES and HOGWILD! on 16 threads for different filtering percentages are shown in Figure 6. Full results of the experiment are presented in the supplemental material. CYCLADES fails to get much speedup when nearly all the features are used. However, as more dense features are removed, CYCLADES obtains a better speedup, almost equalling HOGWILD!'s speedup when $0.048\%$ of the densest features are filtered.

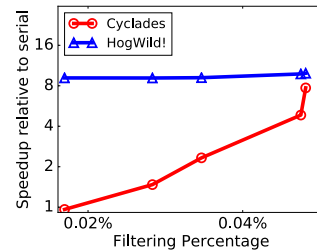

Figure 6: Speedups of CYCLADES and HOGWILD! on 16 threads, for different percentage of dense features filtered. When only a very small number of features are filtered, CYCLADES is almost serial. However, as we increase the percentage from $0.016\%$ to $0.048\%$, the speedup of CYCLADES improves and almost catches up with HOGWILD!.

## 5 Related work

The end of Moore's Law coupled with recent advances in parallel and distributed computing technologies have triggered renewed interest in parallel stochastic optimization [26, 9, 1, 22]. Much of this contemporary work is built upon the foundational work of Bertsekas, Tsitsiklis et al. [3, 23].

Inspired by HOGWILD!'s success at achieving nearly linear speedups for a variety of machine learning tasks, several authors developed other lock-free and asynchronous optimization algorithms, such as parallel stochastic coordinate descent [13]. Additional work in first order optimization and beyond [8, 21, 5], has further demonstrated that linear speedups are generically possible in the asynchronous shared-memory setting.

Other machine learning algorithms that have been parallelized using concurrency control, including non-parametric clustering [18], submodular maximization [19], and correlation clustering [20].

Sparse, graph-based parallel computation are supported by systems like GraphLab [14]. These frameworks require computation to be written in a specific programming model with associative, commutative operations. GraphLab and PowerGraph support serializable execution via locking mechanisms, this is in contrast to our partition-and-allocate coordination which allows us to provide guarantees on speedup.

## 6 Conclusion

We presented CYCLADES, a general framework for lock-free parallelization of stochastic optimization algorithms, while maintaining serial equivalence. Our framework can be used to parallelize a large family of stochastic updates algorithms in a conflict-free manner, thereby ensuring that the parallelized algorithm produces the same result as its serial counterpart. Theoretical properties, such as convergence rates, are therefore preserved by the CYCLADES-parallelized algorithm, and we provide a single unified theoretical analysis that guarantees near linear speedups.

By eliminating conflicts across processors within each batch of updates, CYCLADES is able to avoid all asynchrony errors and conflicts, and leads to better cache locality and cache coherence than HOGWILD!. These features of CYCLADES translate to near linear speedups in practice, where it can outperform HOGWILD!-type of implementations by up to a factor of $5\times$, in terms of speedups.

In the future, we intend to explore hybrids of CYCLADES with HOGWILD!, pushing the boundaries of what is possible in a shared-memory setting. We are also considering solutions for scaling *out* in a distributed setting, where the cost of communication is significantly higher.

## Footnotes

[4] $\widetilde{\Omega}(\cdot)$ and $\widetilde{O}(\cdot)$ hide polylog factors.

[5]Code is available at https://github.com/amplab/cyclades.

[5]It has come to our attention post submission that parts of our partitioning and allocation code could be further parallelized. We refer the reader to our arXiv paper 1605.09721 for the latest results.

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
