[Supplementary Material · cyclades_supplemental.pdf]

# Supplemental Material for: CYCLADES: Conflict-free Asynchronous Machine Learning

## A    Algorithms in the Stochastic Updates family

Here we show that several algorithms belong to the Stochastic Updates (SU) family. These include the well-known stochastic gradient descent, iterative linear solvers, stochastic PCA and others, as well as combinations of weight decay updates, variance reduction methods, and more. Interestingly, even some combinatorial graph algorithms fit under the SU umbrella, such as the maximal independent set, greedy correlation clustering, and others. We visit some of these algorithms below.

**Stochastic Gradient Descent (SGD)**    Given $n$ functions $f_1, \ldots, f_n$, one often wishes to minimize the average of these functions:

$$\min_{\mathbf{x}} \frac{1}{n} \sum_{i=1}^{n} f_i(\mathbf{x}).$$

A popular algorithm to do so —even in the case of non-convex losses— is the stochastic gradient descent:

$$\mathbf{x}_{k+1} = \mathbf{x}_k - \gamma_k \cdot \nabla f_{i_k}(\mathbf{x}_k).$$

In this case, the distribution $\mathcal{D}$ for each sample $i_k$ is usually a with or without replacement uniform sampling among the $n$ functions. For this algorithm the conflict graph between the $n$ possible different updates is completely determined by the support of the gradient vector $\nabla f_{i_k}(\mathbf{x}_k)$.

**Weight decay and regularization**    Similar to SGD, in some cases we might wish to regularize the objective with an $\ell_2$ term and solve instead the following optimization:

$$\min_{\mathbf{x}} \frac{1}{n} \sum_{i=1}^{n} f_i(\mathbf{x}) + \frac{\eta}{2} ||\mathbf{x}||_2^2.$$

In this case, the update is a weighted version of the "non-regularized" SGD:

$$\mathbf{x}_{k+1} = (1 - \gamma_k \eta) \cdot \mathbf{x}_k - \gamma_k \cdot \nabla f_{i_k}(\mathbf{x}_k).$$

The above stochastic update algorithm can be also be written in the SU language. Although here for each single update the entire model has to be updated with the new weights, we show below that with a simple technique it can be equivalently expressed so that each update is sparse and the support is again determined by the gradient vector $\nabla f_{i_k}(\mathbf{x}_k)$.

**First order techniques with variance reduction**    Variance reduction is a technique that is usually employed for (strongly) convex problems, where we wish to minimize the variance of SGD in order to achieve better rates. A popular way to do variance reduction is either through SVRG or SAGA, where a "memory" factor is used in the computation of each gradient update rule. For SAGA we have

$$\mathbf{x}_{k+1} = \mathbf{x}_k - \gamma \cdot \left( \nabla f_{s_k}(\mathbf{x}_k) - \mathbf{g}_{s_k} + \frac{1}{n} \sum_{i=1}^{n} \mathbf{g}_i \right)$$

$$\mathbf{g}_{s_k} = \nabla f_{s_k}(\mathbf{x}_k).$$

For SVRG the update rule is

$$\mathbf{x}_{k+1} = \mathbf{x}_k - \gamma_k \cdot \left( \nabla f_{i_k}(\mathbf{x}_k) - \nabla f_{i_k}(\mathbf{y}) + \mathbf{g} \right)$$

where $\mathbf{g} = \nabla f(\mathbf{y})$, and $\mathbf{y}$ is updated every $T$ iterations of the previous form to be equal to the last $\mathbf{x}_k$ iterate. Again, although at first sight the above updates seem to be dense, we show below how we can equivalently rewrite them so that the update-conflict graph is completely determined by the support of the gradient vector $\nabla f_{i_k}(\mathbf{x}_k)$.

To reiterate, all of the above algorithms, various combinations of them, and further extensions can be written in the language of SU, as presented in Alg. 2.

## A.1 Lazy Updates

For the cases of weight decay/regularization, and variance reduction, we can reinterpret their inherently dense updates in an equivalent sparse form. Let us consider the following generic form of updates:

$$x_j \leftarrow (1 - \mu_j)x_j - \nu_j + h_{ij}(\mathbf{x}_{\mathcal{S}_i}) \tag{1}$$

where $h_{ij}(\mathbf{x}_{\mathcal{S}_i}) = 0$ for all $j \notin \mathcal{S}_i$. Each stochastic update therefore reads from the set $\mathcal{S}_i$ but writes to every coordinate. However, it is possible to make updates lazily only when they are required. Observe that if $\tau_j$ updates are made, each of which have $h_{ij}(\mathbf{x}_{\mathcal{S}_i}) = 0$, then we could rewrite these $\tau_j$ updates in closed form as

$$x_j = (1 - \mu_j)^{\tau_j} x_j - \nu_j \sum_{k=1}^{\tau_j} (1 - \mu_j)^k \tag{2}$$

$$= (1 - \mu_j)^{\tau_j} x_j - \frac{\nu_j}{\mu_j}(1 - \mu_j)\left(1 - (1 - \mu_j)^{\tau_j}\right). \tag{3}$$

This allows the stochastic updates to only write to coordinates in $\mathcal{S}_i$ and defer writes to other coordinates. This procedure is described in Algorithm 3. With CYCLADES it is easy to keep track of $\tau_j$, since we know the serially equivalent order of each stochastic update. On the other hand, it is unclear how a HOGWILD! approach would behave with additional noise in $\tau_j$ due to asynchrony. In fact, HOGWILD! could possibly result in negative values of $\tau_j$, and in practice, we find that it is often useful to threshold $\tau_j$ by $\max(0, \tau_j)$.

**Weight decay and regularization** The weighted decay SGD update is a special case of Eq 1, with $\mu_j = \eta\gamma$ and $\nu_j = 0$. Eq 3 becomes $x_j \leftarrow (1 - \eta\gamma)^{\tau_j} x_j$.

**Variance reduction with sparse gradients** Suppose $\nabla f_i(\mathbf{x})$ is sparse, such that $[\nabla f_i(\mathbf{x})]_j = 0$ for all $\mathbf{x}$ and $j \notin \mathcal{S}_i$. Then we can perform SVRG and SAGA using lazy updates, with $\mu_j = 0$. For SAGA, the update Eq 2 becomes

$$x_j \leftarrow x_j - \gamma\tau_j g_j$$

where $g_j = \left[\frac{1}{n}\sum_{i=1}^{n} \mathbf{y}_{k,i}\right]_j$ is the $j$th coordinate of the average gradient. For SVRG, we instead use $g_j = [\nabla f(\mathbf{y})]_j$.

---

**Algorithm 3** Lazy Stochastic Updates pseudo-algorithm

---

1: **Input:** $\mathbf{x}; f_1, \ldots, f_n; u_1, \ldots, u_n; g_1, \ldots, g_n; \mathcal{D}; T$.
2: Initialize $\rho(j) = 0$.
3: **for** $t = 1 : T$ **do**
4:     sample $i \sim \mathcal{D}$
5:     $\mathbf{x}_{\mathcal{S}_i}$ = read coordinates $\mathcal{S}_i$ from $\mathbf{x}$
6:     **for** $j \in \mathcal{S}_i$ **do**
7:         $\tau_j = t - \rho(j) - 1$.
8:         $x_j \leftarrow (1 - \mu_j)^{\tau_j} x_j - \nu_j \sum_{k=1}^{\tau_j}(1 - \mu_j)^k$.
9:         $x_j \leftarrow (1 - \mu_j)x_j - \nu_j + h_{ij}(\mathbf{x}_{\mathcal{S}_i})$.
10:        $\rho(j) \leftarrow t$.
11: **Output: x**

---

**SGD with dense linear gradients** Suppose instead that the gradient is dense, but has linear form $[\nabla f_i(\mathbf{x})]_j = \lambda_j x_j - \kappa_j + \tilde{h}_{ij}(\mathbf{x}_{\mathcal{S}_i})$, where $\tilde{h}_{ij}(\mathbf{x}_{\mathcal{S}_i}) = 0$ for $j \notin \mathcal{S}_i$. The SGD stochastic update on the $j$th coordinate is then

$$x_j \leftarrow x_j - \gamma(\lambda_j x_j - \kappa_j + \tilde{h}_{ij}(\mathbf{x}_{\mathcal{S}_i}))$$
$$= (1 - \gamma\lambda_j)x_j + \gamma\kappa_j - \gamma\tilde{h}_{ij}(\mathbf{x}_{\mathcal{S}_i}).$$

This fits into our lazy-updates framework with $\mu_j = \gamma\lambda_j$, $\nu_j = -\gamma\kappa_j$, and $h_{ij}(\mathbf{x}_{\mathcal{S}_i}) = -\gamma\tilde{h}_{ij}(\mathbf{x}_{\mathcal{S}_i})$.

**SVRG with dense linear gradients**   Suppose instead that the gradient is dense, but has linear form $[\nabla f_i(\mathbf{x})]_j = \lambda_j x_j - \kappa_j + \tilde{h}_{ij}(\mathbf{x}_{\mathcal{S}_i})$, where $\tilde{h}_{ij}(\mathbf{x}_{\mathcal{S}_i}) = 0$ for $j \notin \mathcal{S}_i$. The SVRG stochastic update on the $j$th coordinate is then

$$x_j \leftarrow x_j - \gamma \left( \lambda_j x_j - \kappa_j + \tilde{h}_{ij}(\mathbf{x}_{\mathcal{S}_i}) - \lambda_j y_j + \kappa_j - \tilde{h}_{ij}(\mathbf{y}_{\mathcal{S}_i}) + g_j \right)$$

$$= (1 - \gamma \lambda_j) x_j - \gamma(g_j - \lambda_j y_j) - \gamma \left( \tilde{h}_{ij}(\mathbf{x}_{\mathcal{S}_i}) - \tilde{h}_{ij}(\mathbf{y}_{\mathcal{S}_i}) \right)$$

where $g_j = [\nabla f(\mathbf{y})]_j$ as above. This fits into our framework with $\mu_j = \gamma \lambda_j$, $\nu_j = \gamma(g_j - \lambda_j y_j)$, and $h_{ij}(\mathbf{x}_{\mathcal{S}_i}) = -\gamma \left( \tilde{h}_{ij}(\mathbf{x}_{\mathcal{S}_i}) - \tilde{h}_{ij}(\mathbf{y}_{\mathcal{S}_i}) \right)$.

# B   With and Without Replacement Proofs

In this Appendix, we show how the sampling and shattering Theorem 2 can be restated for sampling with, or without replacement to establish Theorem 3.

Let us define three sequences of binary random variables $\{X_i\}_{i=1}^n$, $\{Y_i\}_{i=1}^n$, and $\{Z_i\}_{i=1}^n$. $\{X_i\}_{i=1}^n$ consists of $n$ i.i.d. Bernoulli random variables, each with probability $p$. In the second sequence $\{Y_i\}_{i=1}^n$, a random subset of $B$ random variables is set to 1 without replacement. Finally, in the third sequence $\{Z_i\}_{i=1}^n$, we draw $B$ variables with replacement, and we set them to 1. Here, $B$ is integer that satisfies the following bounds

$$(n+1) \cdot p - 1 \leq B < (n+1) \cdot p.$$

Now, consider any function $f$, that has a "monotonicity" property:

$$f(x_1, \ldots, x_i, \ldots, x_n) \geq f(x_1, \ldots, 0, \ldots, x_n), \text{ for all } i = 1, \ldots, n.$$

Let us now define

$$\rho_X = \Pr\left( f(X_1, \ldots, X_n) > C \right)$$
$$\rho_Y = \Pr\left( f(Y_1, \ldots, Y_n) > C \right)$$
$$\rho_Z = \Pr\left( f(Z_1, \ldots, Z_n) > C \right)$$

for some number $C$, and let us further assume that we have an upper bound on the above probability

$$\rho_X \leq \delta.$$

Our goal is to bound $\rho_Y$ and $\rho_Z$. By expanding $\rho_X$ using the law of total probability we have

$$\rho_X = \sum_{b=0}^n \Pr\left( f(X_1, \ldots, X_n) > C \left| \sum_{i=1}^n X_i = b \right. \right) \cdot \Pr\left( \sum_{i=1}^n X_i = b \right) = \sum_{b=0}^n q_b \cdot \Pr\left( \sum_{i=1}^n X_i = b \right)$$

where $q_b = \Pr\left( f(X_1, \ldots, X_n) > C \left| \sum_{i=1}^n X_i = b \right. \right)$, denotes the probability that $f(X_1, \ldots, X_n) > C$ given that a uniformly random subset of $b$ variables was set to 1. Moreover, we have

$$\rho_Y = \sum_{b=0}^n \Pr\left( f(Y_1, \ldots, Y_n) > C \left| \sum_{i=1}^n Y_i = b \right. \right) \cdot \Pr\left( \sum_{i=1}^n Y_i = b \right)$$

$$\overset{(i)}{=} \sum_{b=0}^n q_b \cdot \Pr\left( \sum_{i=1}^n Y_i = b \right)$$

$$\overset{(ii)}{=} q_B \cdot 1 \tag{4}$$

where $(i)$ comes form the fact that $\Pr\left( f(Y_1, \ldots, Y_n) > C \left| \sum_{i=1}^n Y_i = b \right. \right)$ is the same as the probability that that $f(X_1, \ldots, X_n) > C$ given that a uniformly random subset of $b$ variables where set to 1, and $(ii)$ comes from the fact that since we sample without replacement in $Y$, we have that $\sum_i^n Y_i = B$ always.

In the expansion of $\rho_X$, we can keep the $b = B$ term, and lower bound the probability to obtain:

$$\rho_X = \sum_{b=0}^{n} q_b \cdot \Pr\left(\sum_{i=1}^{n} X_i = b\right)$$

$$\geq q_B \cdot \Pr\left(\sum_{i=1}^{n} X_i = B\right) = \rho_Y \cdot \Pr\left(\sum_{i=1}^{n} X_i = B\right) \tag{5}$$

since all terms in the sum are non-negative numbers. Moreover, since $X_i$s are Bernoulli random variables, their sum $\sum_{i=1}^{n} X_i$ is Binomially distributed with parameters $n$ and $p$. We know that the maximum of the Binomial pmf with parameters $n$ and $p$ occurs at $\Pr\left(\sum_i X_i = B\right)$ where $B$ is the integer that satisfies the upper bound mentioned above: $(n+1) \cdot p - 1 \leq B < (n+1) \cdot p$. Furthermore, the maximum value of the Binomial pmf, with parameters $n$ and $p$, cannot be less than the corresponding probability of a uniform element:

$$\Pr\left(\sum_{i=1}^{n} X_i = B\right) \geq \frac{1}{n}. \tag{6}$$

If we combine (5) and (6) we get

$$\rho_X \geq \rho_Y/n \Leftrightarrow \rho_Y \leq n \cdot \delta. \tag{7}$$

The above establish a relation between the without replacement sampling sequence $\{Y_i\}_{i=1}^{n}$, and the i.i.d. uniform sampling sequence $\{X_i\}_{i=1}^{n}$.

Then, for the last sequence $\{Z_i\}_{i=1}^{n}$ we have

$$\rho_Z = \sum_{b=0}^{n} \Pr\left(f(Z_1, \ldots, Z_n) > C \,\middle|\, \sum_{i=1}^{n} Z_i = b\right) \cdot \Pr\left(\sum_{i=1}^{n} Z_i = b\right)$$

$$\overset{(i)}{=} \sum_{b=1}^{B} q_b \cdot \Pr\left(\sum_{i=1}^{n} Z_i = b\right) \tag{8}$$

$$\overset{(ii)}{\leq} \left(\max_{1 \leq b \leq B} q_b\right) \cdot \sum_{b=1}^{B} \Pr\left(\sum_{i=1}^{n} Z_i = b\right)$$

$$\overset{(iii)}{=} q_B = \rho_Y \leq n \cdot \delta,$$

where $(i)$ comes from the fact that $\Pr\left(\sum_{i=1}^{n} Z_i = b\right)$ is zero for $b = 0$ and $b > B$, $(ii)$ comes by applying Hölder's Inequality, and $(iii)$ holds since $f$ is assumed to have the monotonicity property:

$$f(x_1, \ldots, x_i, \ldots, x_n) \geq f(x_1, \ldots, 0, \ldots, x_n),$$

for any sequence of variables $x_1, \ldots, x_n$. Hence, for any $b_1 \geq b_2$

$$\Pr\left(f(Z_1, \ldots, Z_n) > C \,\middle|\, \sum_{i=1}^{n} Z_i = b_1\right) \geq \Pr\left(f(Z_1, \ldots, Z_n) > C \,\middle|\, \sum_{i=1}^{n} Z_i = b_2\right). \tag{9}$$

In conclusion, we have upper bounded $\rho_Z$ and $\rho_Y$ by

$$\rho_Z \leq \rho_Y \leq n \cdot \rho_X \leq n \cdot \delta. \tag{10}$$

**Application to Theorem 3:** For our purposes, the above bound Eq. (10) allows us to assert Theorem 3 for with replacement, without replacement, and i.i.d. sampling, with different constants. Specifically, for any graph $G$, the size of the largest connected component in the sampled subgraph can be expressed as a function $f_G(x_1, \ldots, x_n)$, where each $x_i$ is an indicator for whether the $i$th vertex was chosen in the sampling process. Note that $f_G$ is a monotone function, i.e., $f_G(x_1, \ldots, x_i, \ldots, x_n) \geq f_G(x_1, \ldots, 0, \ldots, x_n)$ since adding vertices to the sampled subgraph may only increase (or keep constant) the size of the largest connected component. We note that the high probability statement of Theorem 2, can be restated so that the constants in front of the size of the connected components accomodate for a statement that is true with probability $1 - 1/n^{\zeta}$, for any constant $\zeta > 1$. This is required to take care of the extra $n$ factor that appears in the bound of Eq. 10, and to obtain Theorem 3.

## C  Robustness against High-degree Outliers

Here, we discuss how CYCLADES can guarantee nearly linear speedups when there is a sublinear $O(n^\delta)$ number of high-conflict updates, as long as the remaining updates have small degree.

Assume that our conflict graph $G_c$ defined between the $n$ update functions has a very high maximum degree $\Delta_o$. However, consider the case where there are only $O(n^\delta)$ nodes that are of that high-degree, while the rest of the vertices have degree much smaller (on the induced subgraph by the latter vertices), say $\Delta$. According to our main analysis, our prescribed batch sizes cannot be greater than $B = (1-\epsilon)\frac{(1-\epsilon)n}{\Delta_o}$. However, if say $\Delta_o = \Theta(n)$, then that would imply that $B = O(1)$, hence there is not room for parallelization by CYCLADES. What we will show, is that by sampling according to $B = (1-\epsilon)\frac{n-O(n^\delta)}{\Delta}$, we can on average expect a parallelization that is similar to the case where the outliers are not present in the conflict graph. For a toy example see Figure 7.

Figure 7: The above conflict graph has a vertex with high degree (i.e., $\Delta_o = 6$), and the remaining of the graph has maximum induced degree $\Delta = 2$. In this toy case, when we sample roughly $\frac{n-1}{\Delta} = 3$ vertices, more often than not, the large degree vertex will not be part of the sampled batch. This implies that when parallelizing with CYCLADES these cases will be as much parallelizable as if the high degree vertex was not part of the graph. Each time we happen to sample a batch that includes the max. degree vertex, then essentially we lose all flexibility to parallelize, and we have to run the serial algorithm. What we establish rigorously is that "on average" the parallelization will be as good as one would hope for even in the case where the outliers are not present.

Our main result for the outlier case follows:

**Lemma 3.** *Let us assume that there are $O(n^\delta)$ outlier vertices in the original conflict graph $G$ with degree at most $\Delta_o$, and let the remaining vertices have degree (induced on the remaining graph) at most $\Delta$. Let the induced update-variable graph on these low degree vertices abide to the same graph assumptions as those of Theorem 1. Moreover, let the batch size be bounded as*

$$B \le \min\left\{(1-\epsilon)\frac{n - O(n^\delta)}{\Delta}, \ O\left(\frac{n^{1-\delta}}{P}\right)\right\}.$$

*Then, the expected runtime of CYCLADES will be $O\left(\frac{E_u \cdot \kappa}{P} \cdot \log^2 n\right)$.*

*Proof.* Let $w_s^i$ denote the total work required for batch $i$ if that batch contains no outlier notes, and $w_o^i$ otherwise. It is not hard to see that $w_s = \sum_i w_s^i = O\left(\frac{E_u \cdot \kappa}{P} \cdot \log^2 n\right)$ and $w_o = \sum_i w_s^i = O\left(E_u \cdot \kappa \cdot \log^2 n\right)$ Hence, the expected computational effort by CYCLADES will be

$$w_s \cdot \Pr\{\text{a random batch contains no outliers}\} + w_o \Pr\{\text{a random batch contains outliers}\}$$

where

$$\Pr\{\text{a random batch contains no outliers}\} = \Omega\left(\left(1 - \frac{1}{n^{1-\delta}}\right)^B\right) \ge 1 - O\left(\frac{B}{n^{1-\delta}}\right) \qquad (11)$$

Hence the expected running time will be proportional to $O\left(\frac{E_u \cdot \kappa}{P} \cdot \log^2 n\right)$, if $O(\frac{E_u \cdot \kappa}{P} \cdot \log^2 n) = O(E_u \cdot \kappa \cdot \log^2 n \cdot \frac{B}{n^{1-\delta}})$, which holds when $B = O\left(\frac{n^{1-\delta}}{P}\right)$. □

## D  Parallel Connected Components Computation

As we will see in the following, the cost of computing CCs in parallel will depend on the number of cores so that uniform allocation across them is possible, and the number of edges that are induced

by the sampled updates on the bipartite update-variable graph $G_u$ is bounded. As a reminder we denote as $G_u^i$ the bipartite subgraphs of the update-variable graph $G_u$, that is induced by the sampled updates of the $i$-th batch. Let us denote as $E_u^i$ the number of edges in $G_u^i$.

Following the sampling recipe of our main text (i.e., sampling each update per batch uniformly and with replacement), let us assume here that we are sampling $c \cdot n$ updates in total, for some constant $c \geq 1$. Assuming that the size of each batch is $B = (1 - \epsilon)\frac{n}{\Delta}$, the total number of sampled batches will be $n_b = \frac{c}{1-\epsilon}\Delta$. The total number of edges in the induced sampled bipartite graphs is a random variable that we denote as

$$Z = \sum_{i=1}^{n_b} E_u^i.$$

Observe that $\mathbb{E}Z = c \cdot E_u$. Using a simple Hoeffding concentration bound we can see that

$$\Pr\{|Z - cE_u| > (1 + \delta)c \cdot E_u\} \leq 2e^{-\frac{2c^2 \cdot (1+\delta)^2 E_u^2}{c \cdot n \Delta_L^2}} \leq 2e^{-2c \cdot (1+\delta)^2 \cdot \frac{n \overline{\Delta}_L^2}{\Delta_L^2}}$$

where $\Delta_L$ is the max left degree of the bipartite graph $G_u$ and $\overline{\Delta}_L$ is its average left degree. Now assuming that

$$\frac{\Delta_L}{\overline{\Delta}_L} \leq \sqrt{n}$$

we obtain

$$\Pr\{|Z - cE_u| > \log n \cdot c \cdot E_u\} \leq 2e^{-c \cdot \log^2 n}.$$

Hence, we get the following simple lemma:

**Lemma 4.** *Let $\frac{\Delta_L}{\overline{\Delta}_L} \leq \sqrt{n}$. Then, the total number of edges $Z = \sum_{i=1}^{n_b} E_u^i$ across the $n_b = \frac{c}{1-\epsilon}\Delta$ sampled subgraphs $G_u^1, \ldots, G_u^{n_b}$ is less than $O(E_u \log n)$ with probability $1 - n^{-\Omega(\log n)}$.*

Now that we have a bound on the number of edges in the sampled subgraphs, we can derive the complexity bounds for computing CCs in parallel. We will break the discussion into the not-too-many- and many-core regime.

**The not-too-many cores regime.** In this case, we sample $n_b$ subgraphs, allocate them across $P$ cores, and let each core compute CCs on its allocated subgraphs using BFS or DFS. Since each batch is of size $B = (1 - \epsilon)\frac{n}{\Delta}$, we need $n_b = \lfloor c \cdot n/B \rfloor = c \cdot \lfloor\frac{\Delta}{1-\epsilon}\rfloor$ batches to cover $c \cdot n$ updates in total. If the number of cores is

$$P = O\left(\frac{\overline{\Delta}_L}{\Delta_L} \cdot \Delta\right),$$

then the max cost of a single CC computation on a subgraph (which is $O(B\Delta_L)$) is smaller than the average cost across $P$ cores, which is $O(Z/P)$. This implies that a uniform allocation is possible, so that $P$ cores can share the computational effort of computing CCs. Hence, we can get the following lemma:

**Lemma 5.** *Let the number of cores be $P = O\left(\frac{\overline{\Delta}_L}{\Delta_L} \cdot \Delta\right)$, and let us sample $O(\Delta)$ batches, where each batch is of size $O(\frac{n}{\Delta})$. Then, each core will not spend more than $O\left(\frac{E_u \log n}{P}\right)$ time in computing CCs, with high probability.*

**The many-cores regime.** When $P >> \frac{\overline{\Delta}_L}{\Delta_L}$ the uniform balancing of the above method will break, leaving no room for further parallelization. In that case, we can use a very simple "push-label" CC algorithm, whose cost on $P$ cores and arbitrary graphs with $E$ edges and max degree $\Delta$ is $O(\max\{\frac{E}{P}, \Delta\} \cdot C_{\max})$, where $C_{\max}$ is the size of the longest-shortest path, or the diameter of the graph. This parallel CC algorithm is given below, where each core is allocated with a number of vertices

The above simple algorithm can be significantly slow for graphs where the longest-shortest path is large. Observe, that in the sampled subgraphs $G_u^i$ the size of the shortest-longest path is always bounded by the size of the largest connected component. By Theorem 3 that is bounded by $O\left(\frac{\log n}{\epsilon^2}\right)$. Hence, we obtain the following lemma.

**Lemma 6.** *For any number of cores $P = O(\frac{n}{\Delta \cdot \Delta_L})$, computing the connected component of a single sampled graph $G_u^i$ can be performed in time $\mathcal{O}(\frac{E_u^i \log n}{P})$, with high probability.*

Since, we are interested in the overall running time for $n_b$ batches of the CC algorithm, we can see that the above lemma simply boils down to the following:

**Corollary 1.** *For any number of cores $P = O(\frac{n}{\Delta \cdot \Delta_L})$, computing the connected component for all sampled graph $G_u^1, \ldots, G_u^{n_b}$ can be performed in time $\mathcal{O}(\frac{E \log^2 n}{P})$.*

**Remark 2.** *In practice it seems to be that parallelizing the CC computation using the not-too-many core regime policy is significantly more scalable.*

---
**Algorithm 5** `push-label`

---
1: Initialize shared $\mathsf{cc}(v)$ variables to vertexIDs
2: **for** $i = 1$ : length of longest shortest path **do**
3:    **for** $v$ in the allocated vertex set **do**
4:       **for** all $u$ that are neighbors of $v$ **do**
5:          Read $cc(v)$ from shared memory
6:          **if** $cc(u) > cc(v)$ **then**
7:             Update shared $cc(u) \leftarrow \min(\mathsf{cc}(u), \mathsf{cc}(v))$

---

# E    Allocating the Conflict Groups

After we have sampled a single batch (i.e., a subgraph $G_u^i$), and computed the CCs for it, we have to allocate the connected components of that sampled subgraph across cores. Observe that each connected component will contain at most $\log n$ updates, each ordered according to the a serial predetermined order. Once a core has been assigned all the CCs, it will process all the updates included in the CCs according to the order that each update has been labeled with.

Now assuming that the cost of the $i$-th update is $w_i$, the cost of a single connected component $\mathcal{C}$ will be $w_{\mathcal{C}} = \sum_{i \in \mathcal{C}} w_i$. We can now allocate the CCs accross cores so that the maximum core load is minimized, using the following $4/3$-approximation algorithm (i.e., an allocation with max load that is at most $4/3$ times the maximum between the max weight, and the sum of weights divided by $P$):To proceed with characterizing the maximum load among the $P$ cores, we assume that the cost of a single update $U_i$ is proportional to the out-degree of that update —according to the update-variable graph $G_u$— times a constant cost which we shall refer to as $\kappa$. Hence, $w_i = O(d_{L,i} \cdot \kappa)$, where $d_{L,i}$ is the degree of the $i$-th left vertex of $G_u$.

Observe that the total cost of computing the updates in a single sampled subgraph $G_u^i$ is proportional to $\mathcal{O}(E_u^i \cdot \kappa)$. Moreover, observe that the maximum weight among all CCs cannot be more than $O(\Delta_L \log n\kappa)$ where $\Delta_L$ is the max left degree of the bipartite update-variable graph $G_u$.

**Lemma 7.** *We can allocate CCs such that the maximum load among cores is $\mathcal{O}\left(\max\left\{\frac{E_u^i}{P}, \Delta_L \log n\right\} \cdot \kappa\right)$, with high probability, where $\kappa$ is the per edge cost for computing one of the $n$ updates defined on $G_u$.*

---
**Algorithm 6** Greedy allocation

---
1: **Input** $\{w_1, \ldots, w_m\}$     % weights to be allocated
2: $b_1 = 0, \ldots, b_P = 0$     % empty buckets
3: **w** = sorted stack of the weights (descending order)
4: **for** $i = 1 : m$ **do**
5:    $w = \mathrm{pop}(\mathbf{w})$
6:    add $w$ to bucket $b_i$ with least sum of weights

---

If $P = O\left(\frac{n}{\Delta \cdot \Delta_L}\right)$ then the average weight will be larger than the maximum divided by a $\log n$ factor, and a near-uniform allocation of CCs according to their weights possible. Since, we are interested in the overall running time for $n_b$ batches, we can see that the above lemma simply boils down to the following:

**Corollary 2.** *For any number of cores $P = O(\frac{n}{\Delta \cdot \Delta_L})$, computing the stochastic updates of the allocated connected component for all sampled graphs (i.e., batches) $G_u^1, \ldots, G_u^{n_b}$ can be performed in time $\mathcal{O}(\frac{E \log^2 n}{P} \cdot \kappa)$.*

| Dataset | # datapoints | # features | Density (average number of features per datapoint) | Comments |
|---|---|---|---|---|
| NH2010 | 48,838 | 48,838 | 4.8026 | Topological graph of 49 Census Blocks in New Hampshire. |
| DBLP | 5,425,964 | 5,425,964 | 3.1880 | Authorship network of 1.4M authors and 4M publications, with 8.65M edges. |
| MovieLens | ~10M | 82,250 | 200 | 10M movie ratings from 71,568 users for 10,682 movies. |
| EN-Wiki | 20,207,156 | 213,272 | 200 | Subset of English Wikipedia dump. |

Table 3: Details of datasets used in our experiments.

| Problem | Algorithm | Dataset | HOGWILD! Stepsize | CYCLADES Stepsize | Batch Size | Average # of connected components | Average size of connected components |
|---|---|---|---|---|---|---|---|
| Least squares | SAGA | NH2010 | $1 \times 10^{-14}$ | $3 \times 10^{-14}$ | 1,000 | 792.98 | 1.257 |
| | | DBLP | $1 \times 10^{-5}$ | $3 \times 10^{-4}$ | 10,000 | 9410.34 | 1.062 |
| Graph eigen | SVRG | NH2010 | $1 \times 10^{-5}$ | $1 \times 10^{-1}$ | 1,000 | 792.98 | 1.257 |
| | | DBLP | $1 \times 10^{-7}$ | $1 \times 10^{-2}$ | 10,000 | 9410.34 | 1.062 |
| Matrix comp | SGD Weighted SGD | MovieLens | $5 \times 10^{-5}$ | | 5,000 | 1663.73 | 3.004 |
| Word embed | SGD | EN-Wiki | $1 \times 10^{-10}$ | | 4,250 | 2571.51 | 1.653 |

Table 4: Stepsizes and batch sizes for the various learning tasks in our evaluation. We selected stepsizes that maximize convergence without diverging. We also chose batch sizes to maximize performance of CYCLADES. We further list the average size of connected components and the average number of connected components in each batch. Typically there are many connected components with small average size, which leads to good load balancing for CYCLADES.

# F  Evaluation — Long version

## F.1  Implementation and Setup

Our experiments were conducted on a machine with 72 CPUs (Intel(R) Xeon(R) CPU E7-8870 v3, 2.10 GHz) on 4 NUMA nodes, each with 18 CPUs, and 1TB of memory. We ran both CYCLADES and HOGWILD! with 1, 4, 8, 16 and 18 threads pinned to CPUs on a single NUMA node (i.e., the maximum physical number of cores possible, for a single node), so that we can avoid well-known cache coherence and scaling issues across different nodes [24]. We note that distributing threads across NUMA nodes significantly increased running times for both CYCLADES and HOGWILD!, but was relatively worse for HOGWILD!. We believe this is due to the poorer locality of HOGWILD!, which results in more cross-node communication. In this paper, we exclusively focus our study and experiments on parallelization within a single NUMA node, and leave cross-NUMA node parallelization for future work, while referring the interested reader to a recent study of the various tradeoffs of ML algorithms on NUMA aware architectures [24].

In Table 3 we list some details of the datasets that we use in our experiments. The stepsizes and batch sizes used for each problem are listed in Table 4, along with dataset and problem details. In general, we chose the stepsizes to maximize convergence without diverging. Batch sizes were picked to optimize performance for CYCLADES.

## F.2  Learning tasks and algorithmic setup

**Least squares via SAGA**   The first problem we consider is least squares:

$$\min_{\mathbf{x}} \frac{1}{n} \|\mathbf{A}\mathbf{x} - \mathbf{b}\|_2^2 = \min_{\mathbf{x}} \frac{1}{n} \sum_{i=1}^{n} (\mathbf{a}_i^T \mathbf{x} - b_i)^2$$

which we will solve using the SAGA algorithm [7], an incremental gradient algorithm with faster than SGD rates on convex, or strongly convex functions. In SAGA, we initialize $\mathbf{g}_i = \nabla f_i(\mathbf{x}_0)$ and iterate the following two steps

$$\mathbf{x}_{k+1} = \mathbf{x}_k - \gamma \cdot \left( \nabla f_{s_k}(\mathbf{x}_k) - \mathbf{g}_{s_k} + \frac{1}{n} \sum_{i=1}^{n} \mathbf{g}_i \right)$$

$$\mathbf{g}_{s_k} = \nabla f_{s_k}(\mathbf{x}_k).$$

where $f_i(\mathbf{x}) = (\mathbf{a}_i^T\mathbf{x} - b_i)^2$ and $\nabla f_i(\mathbf{x}) = 2\left(\mathbf{a}_i^T\mathbf{x} - b_i\right)\mathbf{a}_i$. In the above iteration it is useful to observe that the updates can be performed in a sparse and "lazy" way. That is for any updates where the sampled gradients $\nabla f_{s_k}$ have non-overlapping support, we can still run them in parallel, and apply the vector of gradient sums at the end of a batch "lazily". We explain the details of the lazy updates in Appendix A.1. This requires computing the number of skipped gradient sum updates, say they were $\tau_j$ of them for each lazily updated coordinate $j$, which may be negative in HOGWILD! due to re-ordering of updates. We thresholded $\tau_j$ when needed in the HOGWILD! implementation, as this produced better convergence for HOGWILD!. Unlike other experiments, we used different stepsizes $\gamma$ for CYCLADES and HOGWILD!, as HOGWILD! would often diverge with larger stepsizes. The stepsizes chosen for each were the largest such that the algorithms did not diverge. We used the DBLP and NH2010 datasets for this experiment, and set $\mathbf{A}$ as the adjacency matrix of each graph. For NH2010, the values of $\mathbf{b}$ were set to population living in the Census Block. For DBLP we used synthetic values: we set $\mathbf{b} = \mathbf{A}\tilde{\mathbf{x}} + 0.1\tilde{\mathbf{z}}$, where $\tilde{\mathbf{x}}$ and $\tilde{\mathbf{z}}$ were generated randomly. The SAGA algorithm was run for up to 500 epochs for each dataset.

**Graph eigenvector via SVRG**   Given an adjacency matrix $\mathbf{A}$, the top eigenvector of $\mathbf{A}^T\mathbf{A}$ is useful in several applications such as spectral clustering, principle component analysis, and others. In a recent work, [10] proposes an algorithm for computing the top eigenvector of $\mathbf{A}^T\mathbf{A}$ by running intermediate SVRG steps to approximate the shift-and-invert iteration. Specifically, at each step SVRG is used to solve

$$\min \frac{1}{2}\mathbf{x}^T(\lambda\mathbf{I} - \mathbf{A}^T\mathbf{A})\mathbf{x} - \mathbf{b}^T\mathbf{x} = \min \sum_{i=1}^{n}\left(\frac{1}{2}\mathbf{x}^T\left(\frac{\lambda}{n}\mathbf{I} - \mathbf{a}_i\mathbf{a}_i^T\right)\mathbf{x} - \frac{1}{n}\mathbf{b}^T\mathbf{x}\right).$$

According to [10], if we initialize $\mathbf{y} = \mathbf{x}_0$ and assume $\|\mathbf{a}_i\| = 1$, we have to iterate the following updates

$$\mathbf{x}_{k+1} = \mathbf{x}_k - \gamma \cdot n \cdot \left(\nabla f_{s_k}(\mathbf{x}_k) - \nabla f_{s_k}(\mathbf{y})\right) + \gamma \cdot \nabla f(\mathbf{y})$$

where after every $T$ iterations we update $\mathbf{y} = \mathbf{x}_k$, and the stochastic gradients are of the form $\nabla f_i(\mathbf{x}) = \left(\frac{\lambda}{n}\mathbf{I} - \mathbf{a}_i\mathbf{a}_i^T\right)\mathbf{x} - \frac{1}{n}\mathbf{b}$.

We apply CYCLADES to SVRG with dense linear gradients (see App. A.1) for parallelizing this problem, which uses lazy updates to avoid dense operations on the entire model $\mathbf{x}$. This requires computing the number of skipped updates, $\tau_j$, for each lazily updated coordinate, which may be negative in HOGWILD! due to re-ordering of updates. In our HOGWILD! implementation, we thresholded the bookkeeping variable $\tau_j$ (described in App. A.1), as we found that this produced faster convergence. The rows of $\mathbf{A}$ are normalized by their $\ell_2$-norm, so that we may apply the SVRG algorithm of [10] with uniform sampling. Two graph datasets were used in this experiment. The first, DBLP (http://snap.stanford.edu/data) is an authorship network consisting of 1.4M authors and 4M publications, with 8.65M edges. The second, NH2010 (http://cise.ufl.edu/research/sparse/matrices/DIMACS10/nh2010.html) is a weighted topological graph of 49 Census Blocks in New Hampshire, with an edge between adjacent blocks, for a total of 234K edges. We ran SVRG for 50 and 100 epochs for NH2010 and DBLP respectively.

**Matrix completion via SGD**   In the matrix completion problem, we are given a partially observed $n \times m$ matrix $\mathbf{M}$, and wish to factorize it as $\mathbf{M} \approx \mathbf{UV}$ where $\mathbf{U}$ and $\mathbf{V}$ are low rank matrices with dimensions $n \times r$ and $r \times m$ respectively. This may be achieved by optimizing

$$\min_{\mathbf{U},\mathbf{V}} \sum_{(i,j)\in\Omega} (M_{i,j} - \mathbf{U}_{i,\cdot}\mathbf{V}_{\cdot,j})^2$$

where $\Omega$ is the set of observed entries, which can be approximated by SGD on the observed samples. The objective can also be regularized as:

$$\min_{\mathbf{U},\mathbf{V}} \sum_{(i,j)\in\Omega} (M_{i,j} - \mathbf{U}_{i,\cdot}\mathbf{V}_{\cdot,j})^2 + \frac{\lambda}{2}(\|\mathbf{U}\|_F^2 + \|\mathbf{V}\|_F^2) = \min_{\mathbf{U},\mathbf{V}} \sum_{(i,j)\in\Omega} \left((M_{i,j} - \mathbf{U}_{i,\cdot}\mathbf{V}_{\cdot,j})^2 + \frac{1}{|\Omega|}\frac{\lambda}{2}(\|\mathbf{U}\|_F^2 + \|\mathbf{V}\|_F^2)\right).$$

The regularized objective can be optimized by weighted SGD, which samples $(i,j) \in \Omega$ and updates

$$\mathbf{U}_{i',\cdot} \leftarrow \begin{cases} (1-\gamma\lambda)\mathbf{U}_{i,\cdot} - \gamma \cdot |\Omega| \cdot 2(\mathbf{U}_{i,\cdot}\mathbf{V}_{\cdot,j} - M_{i,j})(\mathbf{V}_{\cdot,j})^T & \text{if } i = i' \\ (1-\gamma\lambda)\mathbf{U}_{i',\cdot} & \text{otherwise} \end{cases}$$

and analogously for $\mathbf{V}_{\cdot,j}$ In our experiments, we chose a rank of $r = 100$, and ran SGD and weighted SGD for 200 epochs. We used the MovieLens 10M dataset containing 10M ratings for 10,000 movies by 72,000 users.

**Word embedding via SGD**   Semantic word embeddings aim to represent the meaning of a word $w$ via a vector $\mathbf{v}_w \in \mathbb{R}^r$. In a recent work by [2], the authors propose using a generative model, and solving for the MLE which is equivalent to:

$$\min_{\{\mathbf{v}_w\},C} \sum_{w,w'} A_{w,w'} (\log(A_{w,w'}) - \|\mathbf{v}_w + \mathbf{v}_{w'}\|_2^2 - C)^2,$$

where $A_{w,w'}$ is the number of times words $w$ and $w'$ co-occur within $\tau$ words in the corpus. In our experiments we set $\tau = 10$ following the suggested recipe of the aforementioned paper. We can approximate the solution to the above problem by SGD: we can repeatedly sample entries $A_{w,w'}$ from $\mathbf{A}$ and update the corresponding vectors $\mathbf{v}_w, \mathbf{v}_{w'}$. In this case the update is of the form as:

$$\mathbf{v}_w = \mathbf{v}_w + 4\gamma A_{w,w'} (\log(A_{w,w'}) - \|\mathbf{v}_w + \mathbf{v}_{w'}\|_2^2 - C)(\mathbf{v}_w + \mathbf{v}_{w'})$$

and identically for $\mathbf{v}_{w'}$ Then, at the end of each full pass over the data, we update the constant $C$ by its locally optimal value, which can be calculated in closed form:

$$C \leftarrow \frac{\sum_{w,w'} A_{w,w'} (\log(A_{w,w'}) - \|\mathbf{v}_w + \mathbf{v}_{w'}\|_2^2)}{\sum_{w,w'} A_{w,w'}}.$$

In our experiments, we optimized for a word embedding of dimension $r = 100$, and tested on a 80MB subset of the English Wikipedia dump available at `http://mattmahoney.net/dc/text.html`. The dataset contains 213K words and $\mathbf{A}$ has 20M non-zero entries. For our experiments, we run SGD for 200 epochs.

### F.3   Speedup and Convergence Results

In this subsection, we present the bulk of our experimental findings.

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

| # threads | Least Squares SAGA NH2010 | Least Squares SAGA DBLP | Graph Eig. SVRG NH2010 | Graph Eig. SVRG DBLP | Mat. Comp. SGD MovieLens | Mat. Comp. Weighted SGD MovieLens | Word2Vec SGD EN-Wiki |
|---|---|---|---|---|---|---|---|
| 1 | 1.9155 | 2.2245 | 0.9039 | 0.9862 | 0.7567 | 0.5507 | 0.5299 |
| 4 | 4.1461 | 4.6099 | 1.6244 | 2.8327 | 1.8832 | 1.4509 | 1.1509 |
| 8 | 6.1157 | 7.6151 | 2.0694 | 4.3836 | 3.2306 | 2.5697 | 1.9372 |
| 16 | 11.7033 | 13.1351 | 3.2559 | 6.2161 | 5.5284 | 4.6015 | 3.5561 |
| 18 | 11.5580 | 14.1792 | 4.7639 | 6.7627 | 6.1663 | 5.5270 | 3.9362 |

Table 5: Cost of partitioning and allocation. The table shows the ratio of the time that CYCLADES consumes for partition and allocation over the time that HOGWILD! takes for 1 full pass over the dataset. On 18 threads, CYCLADES takes between 4-14 HOGWILD! epochs to perform partitioning. Note however, this computational effort is only required once per dataset.

**Stochastic updates running time**  When performing stochastic updates, CYCLADES has better cache locality and coherence, but requires synchronization after each batch. Table 6 shows the time

for each method to complete a single pass over the dataset, only with respect to stochastic updates (i.e., here we factor out the partitioning time). In most cases, CYCLADES is faster than HOGWILD!. In the cases where CYCLADES is not faster, the overheads of synchronizing outweigh the gains from better cache locality and coherency. However, in some of these cases, synchronization can help by preventing errors due to asynchrony that lead to worse convergence, thus allowing CYCLADES to use larger stepsizes and maximize convergence speed.

| # threads | Mat. Comp. SGD MovieLens | | Mat. Comp. $\ell_2$-SGD MovieLens | | Word2Vec SGD EN-Wiki | | Graph Eig. SVRG NH2010 | | Graph Eig. SVRG DBLP | | Least Squares SAGA NH2010 | | Least Squares SAGA DBLP | |
|---|---|---|---|---|---|---|---|---|---|---|---|---|---|---|
| | Cyc | Hog | Cyc | Hog | Cyc | Hog | Cyc | Hog | Cyc | Hog | Cyc | Hog | Cyc | Hog |
| 1 | **2.76** | 2.87 | **3.69** | 3.84 | **9.85** | 10.25 | 0.07 | 0.07 | 11.54 | **11.50** | 0.04 | 0.04 | **5.01** | 5.25 |
| 4 | **1.00** | 1.17 | **1.27** | 1.51 | **2.98** | 3.35 | 0.04 | 0.04 | **4.60** | 4.81 | 0.03 | 0.03 | **1.93** | 1.96 |
| 8 | **0.57** | 0.68 | **0.71** | 0.86 | **1.61** | 1.89 | 0.03 | 0.03 | **2.86** | 3.03 | 0.01 | 0.01 | 1.04 | **1.03** |
| 16 | **0.35** | 0.40 | **0.42** | 0.48 | **0.93** | 1.11 | 0.02 | 0.02 | **2.03** | 2.15 | 0.01 | 0.01 | 0.59 | **0.55** |
| 18 | **0.32** | 0.36 | **0.37** | 0.40 | **0.86** | 1.03 | 0.02 | 0.02 | **1.92** | 2.01 | 0.01 | 0.01 | 0.52 | **0.51** |

Table 6: Time, in seconds, to complete one epoch (i.e. full pass of stochastic updates over the data) by CYCLADES and HOGWILD!. Lower times are highlighted in boldface. CYCLADES is usually faster than HOGWILD!, due to its better cache locality and coherence properties.

## F.5  Diminishing stepsizes

In the previous experiments we used constant stepsizes. Here, we investigate the behavior of CYCLADES and HOGWILD! in the regime where we decrease the stepsize after each epoch. In particular, we ran the matrix completion experiments with SGD (with and without regularization), where we multiplicatively updated the stepsize by 0.95 after each epoch. The convergence and speedup plots are given in Figure 11. CYCLADES is able to achieve a speedup of up to $6 - 7\times$ on $16 - 18$ threads. On the other hand, HOGWILD! is performing worse comparatively to its performance with constant stepsizes (Figure 9(c)). The difference is more significant on regularized SGD, where we have to perform lazy updates (Appendix A.1), and HOGWILD! fails to achieve the same optimum as CYCLADES with multiple threads. Thus, on 18 threads, HOGWILD! obtains a maximum speedup of $3\times$, whereas CYCLADES attains a speedup of $6.5\times$.

(a) Convergence, SGD    (b) Convergence, $\ell_2$-SGD    (c) Speedup, SGD    (d) Speedup, $\ell_2$-SGD

Figure 11: Convergence and speedups for SGD and weighted SGD with diminishing stepsizes for the matrix completion on the MovieLens dataset. In this case, CYCLADES outperforms HOGWILD! by achieving up to 6-7x speedup, when HOGWILD! achieves at most 5x speedup for 16-18 threads. For the weighted SGD algorithm, we used lazy updates (Appendix A.1), in which case HOGWILD! on multiple threads gets to a worse optimum.

## F.6  Binary Classification and Dense Coordinates

In addition to the above experiments, here we explore settings where CYCLADES is expected to perform poorly due to the inherent density of updates (i.e., for data sets with dense features). In particular, we test CYCLADES on a classification problem for text based data, where a few features appear in most data points. Specifically, we run classification for the URL dataset [15] contains $\sim 2.4$M URLs, labeled as either benign or malicious, and 3.2M features, including bag-of-words representation of tokens in the URL.

| Filtering % | # filtered features | # remaining features |
|---|---|---|
| 0.048% | 1,555 | 3,228,887 |
| 0.047% | 1,535 | 3,228,907 |
| 0.034% | 1,120 | 3,229,322 |
| 0.028% | 915 | 3,229,527 |
| 0.016% | 548 | 3,229,894 |

Figure 12: Filtering of features in URL dataset. with a total of 3,230,442 features before filtering. The maximum percentage of features filtered is less than 0.05%.

Figure 13: Convergence and speedups of CYCLADES and HOGWILD! on 1, 4, 8, 16, 18 threads, for different percentage of dense features filtered.

For this classification task, we used a logistic regression model, trained using SGD. By its power-law nature, the dataset consists of a small number of extremely dense features which occur in nearly all updates. Since CYCLADES explicitly avoids all conflicts, for these dense cases it will have a schedule of SGD updates that leads to poor speedups. However, we observe that most conflicts are caused by a small percentage of the densest features. If these features are removed from the dataset, CYCLADES is able to obtain much better speedups. To that end, we ran CYCLADES and HOGWILD! after filtering the densest $0.016\%$ to $0.048\%$ of features. The number of features that are filtered are shown in Table 12.

Full results of the experiment are presented in Figure 13.

# G   Complete Experiment Results

In this section, we present the remaining experimental results that were left out for brevity from our main experimental section. In Figures 14 and 15, we show the convergence behaviour of our algorithms, as a function of the overall time, and then as a function of the time that it takes to perform only the stochastic updates (i.e., in Fig. 15 we factor out the graph partitioning, and allocation time). In Figure 16, we provide the complete set of speedup results for all algorithms and data sets we tested, in terms of the number of cores. In Figure 17, we provide the speedups in terms of the the the computation of the stochastic updates, as a function of the number of cores. Then, in Figures 18 – 21, we present the convergence, and speedups of the overal computation, and then of the stochastic updates part, for our dense feature URL data set. Finally, in Figure 22 we show the divergent behavior of HOGWILD! for the least square experiments with SAGA, on the NH2010 and DBLP datasets. Our overall observations here are similar to the main text. One additional note to make is that when we take a closer look to the figures relative to the times and speedups of the stochastic updates part of CYCLADES (i.e., when we factor out the time of the graph partitioning part), we see that CYCLADES is able to perform stochastic updates faster than HOGWILD! due to its superior spatial and temporal access locality patterns. If the coordination overheads for CYCLADES are excluded, we are able to improve speedups, in some cases by up to 20-70% (Table 7). This suggests that by further optimizing the computation of connected components, we can hope for better overall speedups of CYCLADES.

| | Mat. Comp. SGD MovieLens | Mat. comp $\ell_2$-SGD MovieLens | Word2Vec SGD EN-Wiki | Graph Eig. SVRG NH2010 | Graph Eig. SVRG DBLP | Least Squares SAGA NH2010 | Least Squares SAGA DBLP |
|---|---|---|---|---|---|---|---|
| Overall Speedup | 8.8010 | 7.6876 | 10.4299 | 2.9134 | 4.7927 | 4.4790 | 4.6405 |
| Speedup of Updates | 9.0453 | 7.9226 | 11.4610 | 3.4802 | 5.5533 | 4.6998 | 8.1133 |
| % change | 2.7759% | 3.0580% | 9.8866% | 19.4551% | 15.8718% | 4.9285% | 74.8408% |

Table 7: Speedups of CYCLADES at 16 threads. Two versions speedups are given for each problem: (1) with the overall running time, including the coordination overheads, and (2) using only the running time for stochastic updates. Speedups using only stochastic updates are up to 20% better, which suggests we could potentially observe larger speedups by further optimizing the computation of connected components.

(a) LS, NH2010, SAGA    (b) LS, DBLP, SAGA    (c) Graph Eig., NH2010, SVRG

(d) Graph Eig., DBLP, SVRG

(e) Mat. Comp., 10M, $\ell_2$-SGD    (f) Mat. Comp., 10M, SGD    (g) Word2Vec, EN-Wiki, SGD

Figure 14: Convergence of CYCLADES and HOGWILD! on various problems, using 1, 8, 16 threads, in terms of overall running time. CYCLADES is initially slower, but ultimately reaches convergence faster than HOGWILD!.

Cyclades, 1 threads · · · · Cyclades, 8 threads — Cyclades, 18 threads
HogWild!, 1 threads · · · · HogWild!, 8 threads — HogWild!, 18 threads

(a) LS, NH2010, SAGA  (b) LS, DBLP, SAGA  (c) Graph eigenvector, NH2010, SVRG

(d) Graph eigenvector, DBLP, SVRG

(e) Matrix completion, MovieLens 10M, weighted SGD  (f) Matrix completion, MovieLens 10M, SGD  (g) Word embeddings, EN-Wiki, SGD

Figure 15: Convergence of CYCLADES and HOGWILD! on various problems, using 1, 8, 16 threads, in terms of running time for stochastic updates.

(a) LS, NH2010, SAGA

(b) LS, DBLP, SAGA

(c) Graph Eig., NH2010, SVRG

(d) Graph Eig., DBLP, SVRG

(e) Mat. Comp., 10M, $\ell_2$-SGD

(f) Mat. Comp., 10M, SGD

(g) Word2Vec, EN-Wiki, SGD

Figure 16: Speedup of CYCLADES and HOGWILD! on various problems, using 1, 4, 8, 16 threads, in terms of overall running time. On multiple threads, CYCLADES always reaches $\epsilon$ objective faster than HOGWILD!. In some cases (16(a), 16(e), 16(g)), CYCLADES is faster than HOGWILD! on even 1 thread, as CYCLADES has better cache locality.

(a) LS, NH2010, SAGA

(b) LS, DBLP, SAGA

(c) Graph Eig., NH2010, SVRG

(d) Graph Eig., DBLP, SVRG

(e) Mat. Comp., 10M, $\ell_2$-SGD

(f) Mat. Comp., 10M, SGD

(g) Word2Vec, EN-Wiki, SGD

Figure 17: Speedup of CYCLADES and HOGWILD! on various problems, using 1, 4, 8, 16 threads, in terms of running time for stochastic updates.

Figure 18: Convergence of CYCLADES and HOGWILD! on the malicious URL detection problem, using 1, 4, 8, 16 threads, in terms of overall running time, for different percentage of features filtered.

Figure 19: Convergence of CYCLADES and HOGWILD! on the malicious URL detection problem, in terms of running time for stochastic updates, for different percentage of features filtered.

(a) 0.016%    (b) 0.028%    (c) 0.047%

(d) 0.048%

Figure 20: Speedup of CYCLADES and HOGWILD! on the malicious URL detection problem, using 1, 4, 8, 16 threads, in terms of overall running time, for different percentage of features filtered.

(a) 0.016%    (b) 0.028%    (c) 0.047%

(d) 0.048%

Figure 21: Speedup of CYCLADES and HOGWILD! on the malicious URL detection problem, in terms of running time for stochastic updates, for different percentage of features filtered.

Figure 22: Convergence of CYCLADES and HOGWILD! on least squares using SAGA, with 16 threads, on the NH2010 and DBLP datasets. CYCLADES was able to converge using larger stepsizes, but HOGWILD! often diverged with the same large stepsize. Thus, we were only able to use smaller stepsizes for HOGWILD! in the multi-threaded setting.