[Reviews · NeurIPS 2016]

Reviewer 1

Summary

This paper proposes a general framework for parallelizing stochastic optimization algorithms, namely CYCLADES. This algorithm essentially follows the serial updates, thus needs iterative synchronizations. The asynchronous parallel part is to split workload into multiple independent tasks and assign it to multiple cores and let them work in the asynchronously. From my perspective, the title “asynchronous” is misleading. The proposed algorithm is not comparable to Hogwild! and other asynchronous algorithms. The proposed method is essentially a synchronous algorithm. The key component in the proposed algorithm is the connected component (CC) algorithm, which seems to be an existing algorithm from the paper. I did not find any details about this algorithm or citation. What is the complexity of this algorithm? Can you parallelize it? Do you take account the complexity of running CC algorithm, since this is an additional step to the serial algorithm?

Qualitative Assessment

The clarity of this paper can be improved largely. Some definitions are omitted. Just name a few: 1) What is “d” in line 96. 2) G_c is not well defined in line 77. Are only u_i’s the vertices in this graph?

Confidence in this Review

2-Confident (read it all; understood it all reasonably well)


Reviewer 2

Summary

This is an interesting paper that presents a framework to speedup a family of popular algorithms. The theoretical analysis looks sound and experiments are convincing.

Qualitative Assessment

This is an interesting paper that presents a framework to speedup a family of popular algorithms. The theoretical analysis looks sound and experiments are convincing. - The idea of conflict graph is interesting for the community. - It also appears novel to rewrite SVRG in the SU language.

Confidence in this Review

2-Confident (read it all; understood it all reasonably well)


Reviewer 3

Summary

This paper considers parallel stochastic optimization with shared memory, and the key contribution is a general purpose framework of partitioning the model updates through workers. It is synchronous over batches, and asynchronous during model updates within each batch for a given a partition of updates. This eliminates the possibility of conflicts and makes it free of locks. The method provides can speedup a family of stochastic optimization algorithms while keeping oblivious to the qualitative performance. The analysis is based on the largest size of the connected component if we randomly sample the vertices of a graph. On real datasets with sufficient sparsity, the proposed method outperforms HOGWILD! significantly.

Qualitative Assessment

This paper is very well written and has significantly innovative results. I have thought about graphs like G_u and G_c in the past, but I wasn’t aware of [8]. This paper makes beautiful use of the results in [8] and derives a black-box guarantee for speedup which is nontrivial even based on [8]. It is clearly an important addition to NIPS.

Confidence in this Review

2-Confident (read it all; understood it all reasonably well)


Reviewer 4

Summary

In this paper, the authors propose an enhancement to the HogWild! Framework, which is claimed to maintain serial equivalence (i.e., produces the same outcome as the serial algorithm). Experimental results demonstrate that the proposed method outperforms HogWild! on a few tasks.

Qualitative Assessment

1) I am afraid that the paper contains over-claims. The Cyclades method actually only conditionally achieves the serial equivalence according to Theorem 1. The conflict degree is highly related to the sparsity of the data, and the condition of Theorem 1 cannot be satisfied in many practical scenarios. For example, for deep learning and other dense applications, the allocation algorithm cannot resolve conflicts at all. Even for sparse applications, when the data block is large, or the delay among different local machines is large, the serial equivalence cannot be achieved either. 2) As for the technical contribution of the paper, it seems not that significant over Hogwild!. To some extent, Cyclades adds a data partitioning step to Hogwild, and this data partitioning is done by leveraging existing bipartite allocation algorithms. 3) The authors made some analysis on the speed up achieved by Cyclades. However, speed up is only a system measure, and from the machine learning perspective, we care more about the balance between accuracy and efficiency. The speed up measure is defined with respect to \epsilon objective, which is a little tricky. \epsilon is defines as the smallest error achievable by parallel algorithm (but not by the serial algorithm) – this is unfair to the serial algorithm. According to my experience, in some cases, the parallel algorithm cannot achieve the same smallest error as the serial algorithms do. So, if we define \epsilon as the smallest error by the series algorithm instead, the parallel algorithm will have no speed-up at all. 4) As for the baseline in the experiment, the authors only compare with HogWild!. Given that there have many so many parallelization mechanisms proposed for distributed machine learning these years, only comparing with Hogwild! cannot sufficiently demonstrate the value of the proposed approach. It is necessary to compare with more and stronger baselines. In general, I am not convinced by the rebuttal. According to the authors, serial equivalence is theoretically guranteed, but to me it is guaranteed in a very trivial way. When the data are dense, in order to gurantee the serial equivalence, one may have to use one single core which will make no sense of parallelization. Only if the data is sparse, the partition algorithm can take effect. Furthermore, many parallelization algorithms design for distributed setting can be easily adapted to the shared-memory setting, therefore only comparing with HogWild! seems not convincing enough to me.

Confidence in this Review

3-Expert (read the paper in detail, know the area, quite certain of my opinion)


Reviewer 5

Summary

This paper presents a framework for parallelizing stochastic optimization algorithms in a shared memory setting. It is asynchronous and requires no memory locking during model updates. It uses update partitioning for allocating updates to different processors, and eliminates conflicts across processors within each batch of updates. Interestingly, while allowing lock-free parallelization, it is able to maintain serial equivalence. Experiments on several algorithms show notable improvement.

Qualitative Assessment

The development of parallelizing stochastic optimization algorithms in this paper is quite interesting. By using careful update partitioning for job distribution to allow lock-free parallel computing, it achieves speedup in a parallel, asynchronous, shared memory setting, while maintaining serial equivalence. The presentation is clear, the idea sounds natural, the analysis appears to be solid, and the experimental results look reasonable. There are several suggestions: 1. a discussion on similar ideas of "update partitioning" seems quite necessary, as it appears to be a natural/popular idea, i.e., using cache locality to improve parallelism. 2. a discussion on how general the method developed is interesting. Is it restricted to "stochastic" optimization algorithms? What exactly do you mean by stochastic optimization algorithms? 3. under what situations may this method fail (less effective)? how often we encounter "update locality" in practice? or we can always achieve speedup whatsoever?

Confidence in this Review

2-Confident (read it all; understood it all reasonably well)